# Effects of Milk Urea Nitrogen (MUN) and Climatological Factors on Reproduction Efficiency of Holstein Friesian and Jersey Cows in the Subtropics

**DOI:** 10.3390/ani11113068

**Published:** 2021-10-27

**Authors:** Edward Cottington Webb, Elandri de Bruyn

**Affiliations:** Production Animal Physiology Research Group, Department of Animal Science, University of Pretoria, Private Bag X20, Hatfield 0028, South Africa; elan3db@gmail.com

**Keywords:** dairy, MUN, urea, reproduction, season, subtropics

## Abstract

**Simple Summary:**

Blood urea nitrogen (BUN) affects the reproduction of dairy cows, but it is challenging to measure routinely. Milk urea nitrogen (MUN) is a useful proxy for BUN, but the effects of MUN and climatological factors on the reproduction efficiency of dairy cows in the subtropics, which is characterized by extreme temperature and humidity indices, are lacking. The present study investigated the effects of MUN and climatological factors on the reproduction efficiency of Holstein Friesian and Jersey cows in South Africa. The results confirm that MUN influences the reproduction efficiency of dairy cows in the subtropics. High MUN, relative humidity, and maximum daily temperatures compromise the reproduction of Holstein Friesian cows. Jersey cows have a lower threshold MUN concentration compared to Holstein Friesian cows, but they are not adversely affected by high humidity or temperatures.

**Abstract:**

This study investigated the effects of MUN and climatological factors on the inter calving period (ICP), reproductive performance (RP%), and reproductive index (RI) in Holstein Friesian (*n* = 1177) and Jersey cows (*n* = 3305) in different seasons in the subtropics. Threshold values for MUN on the reproduction of dairy cows in the subtropics remain controversial due to complex environmental interactions, especially with high environmental temperatures. A retrospective analysis was conducted of data obtained from the National Milk Recording scheme of the Agricultural Research Council (ARC) in South Africa. The results confirm that MUN influences the reproduction of dairy cows in the subtropics. MUN concentrations exceeding 18.1 ± 4.28 mg/dL in Holstein Friesian cows and 13.0 ± 4.70 mg/dL in Jersey cows extended the inter calving period (ICP), and decreased RP% and RI. Jersey cows have a lower threshold MUN concentration compared to Holstein Friesian cows, but they are not adversely affected by high humidity or temperatures, while Holstein Friesian cows are.

## 1. Introduction

Conception and reconception rates of high-producing dairy cows remain problematic in modern dairy production systems [1], especially in the subtropics. Apart from the major effects of nutrition and specific nutrients on the reproduction of dairy cows, other risk factors such as negative energy balance, inflammation, and impairment of the immune response have been shown to affect reconception rates in a highly cause-and-effect related way [1]. Dietary energy levels interact with blood urea nitrogen (BUN) and may influence the pregnancy rates of dairy cows [2]. BUN is a major end product of the urea cycle in ruminants, which occurs in the serum or plasma fractions of the blood [3]. High BUN concentrations may indicate inefficient use of dietary nitrogen (N) in the body [2]. Balancing dietary protein and energy ratios may lead to decreased N losses to the environment and increased production efficiency [4]. By holding protein intake constant with increasing levels of energy in the diet, BUN concentrations are likely to be lowered [5].

Since BUN cannot be measured reliably and routinely, milk urea nitrogen (MUN) is generally used as a proxy of BUN because this non-invasive measurement can be easily obtained. MUN correlates significantly with BUN (r = 0.88) [6,7,8], and both provide an estimation of the level of N loss after the absorption of ammonia from the rumen [9]. Additionally, MUN and BUN are used to indicate the protein to energy ratio in diets of healthy dairy cows [5,10]. MUN is used to determine whether a diet contains too much or too little protein [11]. Since urea easily diffuses from the blood into the milk of the cow, the concentration of BUN directly affects MUN concentrations [12]. It follows that N not used for growth or milk protein synthesis is reflected in the MUN-concentration [11]. The current level of MUN recommended for optimal production should be <20mg/dL [13] with a threshold value of 14 mg/dL [8,13].

As the protein to energy ratio increases, so does MUN [14]. Non-protein nitrogen (NPN) accounts for approximately 5% of milk protein and urea makes up almost half of this value [15]. The concentration of MUN depends on the CP, rumen degradable protein (RDP), and rumen undegradable protein (RUP) as well as the protein quality of the diet, but not on its amino acid balance [8,16]. Conception rates of first service dairy cows are negatively associated with milk production and MUN [7]. High concentrations of MUN at the time of insemination is a contributing factor to higher risk pregnancy failures [17].

The results are contradictory with regard to the effect of breed on MUN concentrations. One study reported a higher MUN concentration in Holstein Friesian cows than in Jersey cows [18]. Another study reported a higher test-day MUN concentration in Jersey compared to Holstein Friesian cows, but this was dependent on whether the measurement was from a single-breed herd or multiple breed herds [19]. In a study where differences were observed between breeds for milk volume, milk protein, and milk fat, no differences in either MUN or BUN concentrations were detected [12].

Seasonal variations in MUN influence cow reproduction (e.g., cows mated in winter with low MUN concentrations were more likely to conceive than cows with a higher MUN, or those mated in the summer [20]). The negative interaction between heat stress and a higher MUN concentration reduces the reproduction potential of dairy cows. Heat stress adversely affects the normal biological processes of the cow, thereby increasing the amount of energy needed to convert ammonia to urea [20]. The concentration of MUN is affected by the month of the year, with lower concentrations (11.8 mg/dL) observed in winter and higher concentrations (18.1 mg/dL) in summer and spring [4]. Other researchers have found that MUN concentrations differ between seasons, namely with higher MUN in summer [19,21,22,23,24]. Threshold values for MUN and the related effects on the reproduction of dairy cows in the subtropics are not well documented and somewhat controversial. Extreme temperature and humidity indices occur in South Africa [25], so the objective of this study was to investigate the influence of MUN and season on the reproduction of Holstein Friesian and Jersey cows in this region.

## 2. Materials and Methods

The Animal Ethics committee of the University of Pretoria (EC171101-156) approved this research. Data consisting of ca. 10,000 records were collected from two Holstein Friesian and two Jersey herds in the Gauteng region of South Africa from 2006 to 2008, during which time no extreme climatic conditions occurred. Dairy farms were selected based on (1) their participation in the National Milk Recording scheme of the Agricultural Research Council (ARC) in South Africa; (2) on the basis of being representative of typical commercial dairy farms using total mixed rations (TMRs) of which the nutritional data were available for comparative purposes; and (3) being close to an official South African weather station to obtain accurate daily weather data. Weather data consisted of daily maximum (T_x_), minimum (T_n_), and average (T_Av_) temperatures, and minimum (RH_n_) maximum (RH_x_) and average humidity RH_Av_), which were calculated from the hourly data. The cows included in this retrospective analysis were representative of at least one complete lactation cycle and in 1st, 2nd, and or 3rd parity with monthly composite milk samples (e.g., complete monthly milk composition data). Records of a total of 1177 Holstein Friesian and 3305 Jersey cows that complied with the criteria described above were included in the analyses.

Seasons were defined as summer (December–February), autumn (March to May), winter (June to August), and spring (September to November). Composite milk samples were analyzed by Mérieux Nutrisciences South Africa (accredited with SANAS, https://www.sanas.co.za/ (accessed on 10 June 2021)); and International Committee for Animal Recording (ICAR), (https://www.icar.org/ (accessed on 10 June 2021)) for butterfat %, protein %, lactose %, milk urea nitrogen (mg/dL), and somatic cell count (cells/mL) (https://www.merieuxnutrisciences.com/za/raw-milk-testing-sa (accessed on 10 June 2021)). The data were captured in Microsoft Excel and checked for accuracy before statistical analyses. The target variables selected were the reproductive characteristics, namely inter calving period (ICP), reproductive performance (RP%), and reproductive index (RI). Inter calving period was calculated as the number of days between consecutive calving dates. Reproductive performance (RP%) was calculated to describe the reproduction success of the dairy cows. It was calculated as follows:RP% = [278/ICP] × 100(1)

The average gestation period of a dairy cow was taken as 278 days based on published data [26,27]. Reproductive index (RI) was calculated as previously described [28], namely:RI = 200 − [Age(D)/[966 + [417 × (Calving number − 1)]] × 100](2)

Age(D) is the age at last calving in days, which was calculated by using the average number of calvings (also referred to as calving number in the dataset obtained from the milk recording scheme) of the cow and correlating it with the breed and then calculating the average age of the cow in months (Age(M)) as described in [29]. Age in months was converted to the average age in days (Age(D)), which was used to calculate RI (Table 1).

RP% and RI were used to obtain more accurate and practical estimates of reproduction efficiency. Summary statistics and frequency distributions were calculated and the data were checked for missing values and outliers. The normality of variables was tested by plotting histograms and employing the Shapiro–Wilk test in IBM SPSS statistics version 27. The effects of fixed effects, random factors, and repeated measures on the target variables were analyzed using the general linearized mixed models analysis procedure in SPSS version 27, with breed (Holstein Friesian or Jersey), season (summer, autumn, winter, spring), parity (1st, 2nd, and or 3rd), and climatological variables as fixed factors and dairy herd and year (2006–2008) as random factors, and including milk composition data as repeated variables. Days in lactation were used as a covariate to correct for the stage of lactation because milk recording data were obtained monthly with cows in different stages of lactation at every recording. Records of animals with incomplete data were omitted, which yielded a final usable dataset of 8110 cow observations in this study. The linearized mixed models analysis procedure was also repeated within breeds due to the known differences between the Holstein Friesian and Jersey breeds in terms of certain milk production characteristics and their genetic differences in terms of adaptability in warm and humid environments by including herd as a random factor and using the same fixed factors and repeated measures as described above.

The effects of main factors, random factors, and repeated measures on reproduction parameters were tested using the significance of coefficients in the general linearized mixed models procedure at *p <* 0.05 and *p <* 0.01. Variance components were estimated using maximum likelihood (ML) and restricted maximum likelihood (REML) methods, and the relationships among variables were estimated through Pearson product-moment correlations (r^2^). A difference was considered significant at a level of *p* ≤ 0.05 and *p* < 0.01.

## 3. Results

### 3.1. Climatological Conditions of the Region

The climatological conditions in the Gauteng region during the period of the study for Holstein Friesian and Jersey herds are presented in Table 2.

The climatological data indicate that the dairy herds sampled are located in a subtropical region. The climate in Gauteng reflects warm and high humidity (relative humidity (RH_Av_ > 50%) in spring and summer, mild conditions in autumn, and about six weeks of cold to very cold and dry (RH_Av_ < 30%) conditions in winter.

### 3.2. Effects of Breed on Milk Production Parameters

Nutritional, milk production, and reproduction parameters for Holstein Friesian and Jersey cows in the study are presented in Table 3. The average crude protein (CP) content of total mixed rations (TMR) fed to dairy cows during the period of this review was 17.1 ± 1.1%, which was 1% higher compared to the 16% reported in a previous study [8], but the average metabolizable energy (ME) content of the diets recorded was similar. The composition of TMRs used in the present study reflects the emphasis on milk yield in Holstein Friesian cows and butterfat production in Jersey cows (e.g., the ME-content was similar). The CP content of TMRs fed to Jersey cows was on average 0.5% higher than that fed to Holstein Friesian cows, but the latter contained ca. 8% more rumen degradable protein (RDP), which increased ruminal N metabolism.

The results from the mixed models analysis confirmed the known difference between Holstein Friesian and Jersey cows in terms of total daily milk yield (*p* < 0.01). This analysis revealed a coefficient of 8.04 (std. error = 1.152), indicating that the average daily milk yield of Holstein Friesian cows was 8,04 kg/day higher (*p* < 0.01) compared to Jersey cows. The butterfat content of milk was higher in composite milk samples from Jersey cows compared to Holstein Friesian cows, which agrees with previous results [9,29]. The Jersey breed is known to produce milk with a higher butterfat content, which is preferred for the production of butter. In contrast, the milk protein content was higher (*p* < 0.01) in samples from Holstein Friesian cows compared to those from Jersey cows, which agrees with previous results [9,29], while the lactose content did not differ between breeds.

A fair assumption was made about somatic cell counts and udder health based on previous research findings [30]. The average somatic cell count recorded in this study (674.6 ± 1395.4 cells/mL) was lower than the proposed threshold of 150,000 cells/mL in composite milk samples [31]. Values exceeding 150,000 cells/mL indicate possible intramammary infection or mastitis, so cows included in the present study were relatively unaffected by intramammary infections or mastitis and therefore had good udder health. The average MUN concentration of all cows pooled was 13.5 ± 5.32 mg/dL, which agrees with previous findings [32,33]. In a similar study in the USA, average MUN concentrations in dairy cows was 12.6 ± 4.0 mg/dL (e.g., 11.1 mg/dL for cows fed a TMR containing a low CP and 13.6 mg/dL for cows fed a TMR containing a moderate CP) [32]. Although the MUN concentrations differed numerically between composite milk samples from Holstein Friesian (18.1 ± 4.28 mg/dL) compared to Jersey cows (13.0 ± 4.70 mg/dL), the difference was not statistically significant (*p* = 0.093) due to other significant fixed and residual effects (*p* < 0.01; Z = 62.658). In Holstein Friesian cows, it was reported that a MUN <14 mg/dL may indicate an insufficient dietary protein intake [33].

In milk samples from both breeds, MUN was influenced by the average daily milk yield, which was associated with an increase in MUN by 0.016 mg/dL per 1 L increase in milk yield (*p* < 0.04) in Jersey cows and 0.039 mg/dL per 1 L increase in milk yield in Holstein Friesian cows (*p* < 0.04). In contrast, MUN decreased by 1.056 mg/dL per 1 MJ increase in ME content. RDP influenced MUN concentrations (*p* < 0.05) by a 0.051 mg/dL increase in MUN per 1% increase in RDP%. Parity also influenced MUN, with milk samples from cows in the 1st parity containing 0.54 mg/dL more MUN (*p* < 0.01) compared to samples from the 2nd and 3rd parities. In Holstein Friesian cows, the effect of parity was more pronounced than in Jersey cows, with cows in the 1st parity containing 2.145 mg/dL more MUN (*p* < 0.01) than samples from cows in the 2nd or 3rd parity.

### 3.3. Reproduction Parameters of Holstein Friesian and Jersey Cows

The average calving number of Holstein Friesian and Jersey cows sampled in the present study was 2.23 ± 0.81, which indicates that the herds sampled contained numerically more cows in the 2nd (*n* = 2352) and 3rd (*n* = 3660) parities, compared to the 1st parity (*n* = 1854) (Table 4).

Breed did not significantly affect the inter calving period (ICP) of dairy cows in this study and herd was not a significant random factor in the variance components model. ICP was influenced by season, parity, MUN, RDP, CP, and ME (Table 5). Cows that calved in summer and spring had 41.7 and 16.2 days shorter ICP respectively, compared to those that calved in winter, while the ICPs did not differ between cows that calved in spring and autumn. The ICP of cows in the 1st, 2nd, and 3rd parity differed significantly among each other. Cows in the 1st and 2nd parity had a 42.2 day and 18.8 day shorter ICP (*p* < 0.01), respectively, compared to those in the 3rd parity. MUN increased the ICP by 1.03 days per 1 mg/dL exceeding 14.3 mg/dL threshold (*p* < 0.01), while CP increased ICP by 4.6 days per 1% increase in CP exceeding 17.1% in dairy cows on average (*p* < 0.01). In contrast, ICP was shortened by 5.5 days per MJ ME exceeding 10.9 MJ ME. The inclusion of climatological variables in the mixed models analysis indicates that ICP is extended by 2.87 days with each 1 °C increase above an average daily temperature of 17.5 °C and extended by 0.94 days for each 1% increase in daily RH_Av_ exceeding 53.14% (*p* < 0.01). In previous studies on dairy cattle in the subtropics, it was indicated that high environmental temperatures and humidity adversely affected their reproduction [34,35]. Heat stress was noted at daily temperatures >23.8 °C and a temperature-humidity index (THI) of 80% [34].

Similar to ICP, reproduction performance (RP%) did not differ between breeds but was influenced by parity, season, MUN, CP%, ME, and climatological factors as outlined in Table 6. Cows in their 1st and 2nd parity had a 5.6% and 2.5% higher RP%, respectively, compared to those in the 3rd parity (*p* < 0.01). MUN and CP decreased the RP% of dairy cows by 0.1% per 1 mg/dL increase in MUN exceeding 14.3 mg/dL and 0.81 % per 1% increase in CP exceeding 17.1% (*p* < 0.01), respectively. However, RDP and ME improved RP% by 0.37% per 1% increase in RDP exceeding 56.4% (*p* < 0.01) and 0.85% per 1 MJ increase in ME exceeding 10.9 MJ in the TMR (*p* < 0.05), respectively. Climatological factors adversely affected RP%, with a 0.42% decrease in RP% per 1 °C increase in average daily temperature exceeding 17.5 °C and 0.11% decrease in RP% per 1% increase in RH_Av_ exceeding 53.1% (*p* < 0.01).

Reproduction index (RI) is a longer-term measure of reproduction efficiency since it considers parity. RI was influenced by the fixed effects, namely breed (*p* < 0.01), seasons (*p* < 0.05), and parity (*p* < 0.01). The RI of Jersey cows was 4.9 units higher (*p* < 0.01) compared to that recorded for Holstein Friesian cows in this subtropical region of South Africa. RI was higher for cows that calved in summer compared to winter, spring, or autumn (*p* < 0.01), while the RI of cows that calved in spring, winter, and autumn did not differ. RI was higher for the 1st parity than the 2nd parity (*p* < 0.01), and lowest in the 3rd parity cows (*p* < 0.01). Climatological factors did not affect RI, but RDP and ME improved RI (*p* < 0.01) by 0.04 and 0.1 units each, respectively. This finding confirms the long-term benefits of improved nutrition on reproduction success as measured over the first three parities in dairy cows.

### 3.4. Effects of Season and Climatological Factors on Reproduction Parameters within Breeds

The effects of season on reproduction parameters within the Holstein Friesian and Jersey breeds are presented in Table 7. The random covariance effect of herds was not statistically significant in the mixed models analysis (Z = 0.482; *p* = 0.630).

#### 3.4.1. Holstein Friesian Cows

The ICP of Holstein Friesian cows was affected most by season (*p* < 0.01), parity (*p* < 0.01), total milk yield (*p* < 0.01), MUN (*p* < 0.05), CP (*p* < 0.01), ME (*p* < 0.05), and RH_Av_ (*p* < 0.01). The ICP of Holstein Friesian cows that calved in summer was shorter compared to those that calved in other seasons, and those that calved in winter was longer (*p* < 0.01). ICP was respectively 39 and 28 days shorter in the 1st and 2nd parity cows compared to those in the 3rd parity (*p* < 0.01). An increase in daily milk yield and MUN extended the ICP by respectively 0.9 days (*p* < 0.01) and 0.8 days (*p* < 0.05). In contrast, ICP decreased by 11.8 days and 7.9 days respectively per unit increase in dietary CP and ME (*p* < 0.01). ICP was extended by 1.44 days per 1% increase in daily RH_Av_ exceeding 54.21% (*p* < 0.01). The effects of daily minimum, maximum, or average temperatures on the ICP of Holstein Friesian cows were not significant.

The RP% of Holstein Friesian cows that calved in summer was respectively 5.2% 2.6%, and 2.4% higher (*p* < 0.01) compared to cows that calved in winter, spring, or autumn. There was no difference between spring and autumn, but both were higher than in winter. RP% was 5.1% higher for 1st compared to 2nd parity cows, and 3.7% higher for 2nd compared to 3rd parity cows (*p* < 0.01). Similar to ICP, RP% was lower (0.13% per 1 L increase in milk yield; *p* < 0.01) and 0.12% lower (*p* < 0.05) per 1 mg/dL increase in MUN. Of the climatological factors, only RH_Av_ was significant, which caused a decrease of 0.18% per 1% increase in daily RH_Av_ exceeding 54.2%. The effects of RI responded similarly to RP%, but were also affected by environmental temperatures, notably, RI decreased by 0.02% per 1 °C increase in maximum daily temperature exceeding 24.8 °C.

#### 3.4.2. Jersey Cows

Seasonal effects were less significant in Jersey cows in this study, with the ICP being shorter only for cows calving in summer compared to the other seasons (*p* < 0.01). ICP was respectively 41.1 and 13.2 days shorter in the 1st and 2nd parity cows compared to those in the 3rd parity (*p* < 0.01). An increase in daily milk yield and MUN extended ICP in Jersey cows by respectively 0.7 and 1,0 days (*p* < 0.01). Increasing RDP above 54.3% decreased ICP by 4.5 days (*p* < 0.01). Jersey cows did not respond significantly to RH_Av_, but ICP was extended by 3.1 days for each 1 °C decrease in minimum daily temperature (T_n_) below 9.4 °C.

RP% responded in a similar way as ICP in Jersey cows, indicating a small but significant increase in reproduction performance for cows calving in summer compared to other seasons (*p* < 0.01). RP% decreased by 5.4% from 1st to 2nd and 1.8% from 2nd to 3rd parity in Jersey cows. RP% decreased by 0.11% per 1 L increase in average daily milk yield (*p* < 0.01), but was unaffected by MUN concentration. RI was not influenced by season in Jersey cows. RH_Av_ did not affect RP% or RI, but minimum daily temperatures decreased RP% in Jersey cows by 0.54% per 1 °C below 9.4 °C (*p* < 0,01).

## 4. Discussion

Data from this study confirm the known differences in milk production and composition between Holstein Friesian and Jersey cows. Holstein Friesian cows had higher daily milk yields and the composite milk samples contained a lower butterfat content compared to that of Jersey cows, which agrees with previous results [9,29]. The composition of the TMRs fed to Holstein Friesian and Jersey cows was comparable to that generally fed to dairy cows in this subtropical region of South Africa and that used in other studies [32,33]. In general, the TMRs contained a comparable CP and ME content, but with a higher percentage of RDP in diets fed to Holstein Friesian cows compared to Jersey cows. This increased the readily available concentration of protein and N in the rumen of Holstein Friesian cows. RDP is broken down to ammonia in the rumen and diffuses into the portal system where it is converted to urea by the liver and converted to BUN [36]. Consequently, the MUN content of composite milk samples from Holstein Friesian cows (18.0 ± 4.28 mg/dL) was numerically higher compared to that of Jersey cows (13,0 ± 4,70 mg/dL), although not statistically significant. The results from the present study indicate that nutrition, through effects on MUN influenced the ICP and reproduction efficiency of dairy cows. These findings agree with a previous study on conception rates in dairy cows [7], which showed lower conception rates for cows with higher concentrations of MUN. In a more recent study, it was emphasized that a high MUN concentration had a particularly negative effect on reproductive success if the MUN concentration was high on the day of insemination [17]. A decrease in the rumen degradable protein (RDP) fraction was shown to improve the rumen microbiome and decreased MUN, which improved the reproduction of cows [5].

On average, MUN in composite milk samples from dairy cows increased by 0.051 mg/dL per 1% increase in RDP, and decreased by 1.056% per 1 MJ ME increase in dietary energy level. It follows that CP, RDP, and ME have to be carefully balanced to prevent excessive MUN concentrations that may adversely affect the reproduction of dairy cows. A MUN concentration <14mg/dL may indicate that the protein content of the diet is insufficient [33]. In a study in Israel in Jersey cows [4], the pregnancy rates of cows increased when the MUN concentration was <11.75 mg/dL. Similar results were reported in Holstein Friesian cow herds in Ohio (USA) with higher pregnancy rates recorded when MUN was <10 mg/dL, with lowered pregnancy rates at MUN concentration >15.4 mg/dL [32]. BUN decreases uterine pH, which is detrimental to embryo survival [37,38,39]. Urea is a water-soluble molecule that easily diffuses into oocytes before ovulation, therefore compromising oocyte quality and cleavage rates of blastocysts after fertilization [13,36].

The effects of MUN and climatological factors on reproduction parameters of Holstein Friesian and Jersey cows in the subtropics were investigated. The Holstein Friesian and Jersey cows included in this study had a similar calving number (2.23 ± 0.81) and reproduction parameters in both breeds were influenced to a varying degree by the fixed factors parity, season, daily milk yield, MUN, RDP, ME, and climatological factors. In both breeds, ICP was extended by an increase in daily milk yield as well as an increase in the MUN content of milk. In Holstein Friesian cows, the ICP was extended by 0.8 days (*p* < 0.05) per 1 mg/dL increase in MUN exceeding 14.3 mg/dL. In contrast, ICP was shortened by 11.8 days and 7.9 days, respectively, per unit increase in dietary CP and ME (*p* < 0.01). The average MUN concentration of Holstein Friesian cows in the present study was less than the proposed threshold value, namely <20 mg/dL for dairy cows previously proposed [13].

However, during summer, Holstein Friesian cows frequently experience heat stress in many regions in South Africa. ICP was extended by 1.44 days per 1% increase in daily RH_Av_ (*p* < 0.01), while the effects of daily minimum, maximum, and average temperatures on the ICP of Holstein Friesian cows were not significant. RH caused a decrease of 0.18% in RP% per 1% increase in daily RH_Av_ exceeding 54.2%. RI in Holstein Friesian cows also exhibited a significant decrease of 0.02% per 1 °C increase in maximum daily temperature exceeding 24.8 °C. Other researchers have also reported seasonal effects on conception rates in dairy cows [20]. Heat stress may burden the animal more by using energy to convert ammonia to urea, instead of thermoregulation.

In the subtropics, heat stress occurs in dairy cows when temperatures exceed 23.8 °C [34]. A decrease of 20% to 30% in conception and pregnancy rates was reported in previous studies during summer [35]. The reduction in pregnancy rate of 13% at 90 days after calving was associated with an increase in the period from calving to fertilization by 13 days. Heat stress may therefore worsen the effect of MUN on reproduction by making it more difficult for a cow to lactate, conceive, and maintain homeostasis. Too much dietary protein in the hot humid seasons may cause rising concentrations of MUN, which, may adversely affect reproduction and fertility in a herd. Such observations have been reported that confirms that a higher MUN concentration in the summer resulted in a reduced reproduction rate in Iranian dairy cows [24].

The ICP, RP%, and RI were less affected by season or climatological factors in Jersey cows. Similar to Holstein Friesian cows, Jersey cows showed an increase in ICP and decrease in RP% and RI in cows with a higher daily milk yield and MUN content of composite milk samples. However, increasing RDP above 54.3% decreased ICP in Jersey cows by 4.5 days. RH and high maximum temperatures did not adversely affect the reproduction parameters of Jersey cows. In contrast to Holstein Friesian cows, Jersey cows were more sensitive to cold rather than warm conditions, for example, the ICP of Jersey cows was extended by 3.1 days for each 1 °C decrease in minimum daily temperature below 9.4 °C.

It follows that MUN may significantly affect the reproduction efficiency of Holstein Friesian and Jersey cows as the concentration of MUN increases above threshold values, as demonstrated by the significant effect of MUN on ICP between and within breeds. These breeds responded adversely to MUN concentrations exceeding the threshold values of 18.1 mg/dL in Holstein Friesian and 12.9 mg/dL in Jersey cows. Since the negative effect of MUN on the ICP of Jersey cows was observed at MUN concentrations below 14 mg/dL, it follows that Jersey cows may have a lower threshold for MUN in terms of possible adverse effects on reproduction efficiency compared to Holstein Friesian cows. It follows that both protein inclusion, RDP, and the derivative, MUN, should be considered in diet formulations for Jersey cows.

The reproduction performance of Holstein Friesian cows was affected more by climatological factors such as the relative humidity (RH_Av_ > 54.3%) and maximum daily temperatures (T_X_ > 24.8 °C), while Jersey cows were sensitive to minimum temperatures decreasing below 9.4 °C. These findings suggest different MUN threshold values for different dairy cattle breeds in different seasons. Jersey cows are generally smaller, better adapted to the subtropics, and have a lower genetic capacity for milk production than Holstein Friesian cows. This may explain the variation in responsiveness of dairy cows to RDP and negative effects on reproduction parameters due to its conversion to BUN. Approximately 12.54 g of urinary nitrogen is excreted per unit of MUN [40,41]. More urinary nitrogen is excreted per unit of MUN than previously described [12] and as CP increased, the efficiency of N-excretion in the milk decreased, while urinary excretion increased. The efficiency of N-excretion in the milk decreased from 52,3% in diets containing 13% CP to 42.7% in diets containing 17% CP.

The findings of this study have several important implications for future practice. The use of MUN to manage reproduction in dairy cattle differs between breeds and seasons, while the effects of dietary protein, RDP, and interactions with ME and climatological factors on MUN are important.

## 5. Conclusions

Sustainable strategies to reduce heat stress and manage MUN concentrations in dairy cows are very important to improve reproduction efficiency in the subtropics. The effect of MUN on the reproduction efficiency of dairy cows is significant in Holstein Friesian and Jersey cows in the subtropics, but their susceptibility to excessive MUN concentrations depend on climatological factors such as minimum and maximum daily temperatures and relative humidity. The MUN threshold values obtained in this subtropical environment were 18.1 mg/dL for Holstein Friesian and 12.9 mg/dL for Jersey cows. Holstein Friesian and Jersey cows took longer to conceive as the concentration of MUN increased above these threshold values. Reproduction performance of Holstein Friesian cows was compromised by high relative humidity (>54.3%) and high maximum temperatures (>24.8 °C), while Jersey cows were adversely affected by minimum daily temperatures below 9.4 °C.

## Figures and Tables

**Table 1 animals-11-03068-t001:** Calculation of average cow age (in days), according to breed and calving number, used in the calculation of RI.

Breed	Calving Number	Age(M) *	Age(D)
Jersey	1	28	852
	2	41	1247
	3	54	1643
	4	82	2494
Holstein Friesian	1	29	882
	2	42	1278
	3	58	1764
	4	90	2738

* As previously calculated [29].

**Table 2 animals-11-03068-t002:** Seasonal climatological conditions during the period of the study in the Gauteng region where data were obtained for Holstein Friesian and Jersey dairy herds.

Breed	Season	T_n_ (°C)	T_x_ (°C)	T_Av_ (°C)	RH_Av_ (%)
Holstein Friesian	Spring	13.4 ± 1.58	26.1 ± 1.51	19.8 ± 1.26	52.8 ± 12.47
	Summer	16.7 ± 0.65	28.6 ± 2.18	22.6 ± 1.24	61.8 ± 8.39
	Autumn	11.5 ± 3.53	25.1 ± 3.20	18.3 ± 3.23	54.5 ± 8.79
	Winter	6.2 ± 1.78	20.6 ± 1.66	13.4 ± 1.57	49.7 ± 7.41
	Average	11.8 ± 4.39	24.8 ± 3.63	18.3 ± 3.91	54.2 ± 10.63
Jersey	Spring	11.6 ± 2.57	26.9 ± 1.84	19.3 ± 1,58	50.0 ± 10.58
	Summer	15.7 ± 1.56	28.3 ± 1.44	22.1 ± 0.96	60.9 ± 4.60
	Autumn	10.0 ± 4.22	24.2 ± 2.32	17.1 ± 3.12	56.5 ± 6.94
	Winter	2.5 ± 2.45	21.1 ± 1.89	11.8 ± 1.69	47.4 ± 6.74
	Average	9.5 ± 5.68	25.0 ± 3.44	17.2 ± 4.34	52.7 ± 9.41

T_n_—Minimum daily temperature; T_x_—Maximum daily temperature; T_Av_—Average daily temperature; RH_Av_—Average relative humidity.

**Table 3 animals-11-03068-t003:** Summary statistics (mean ± SD) of milk production parameters in Holstein Friesian and Jersey cows included in this study.

	Holstein Friesian (*n* = 1177)	Jersey (*n* = 3305)
**TMR-Nutritional Parameters ^#^**		
Crude protein [%]	16.6 ± 0.82	17.3 ± 1.11
Rumen degradable protein [%]	62.2 ± 1.23	54.4 ± 2.42
Metabolizable energy [MJ/kg]	10.9 ± 0.67	10.9 ± 0.53
**Milk Production Parameters**		
Average milk yield [kg/d]	30.1 ± 8.62 ^a^	21.7 ± 7.58 ^b^
Butterfat %	4.0 ± 0.59 ^a^	4.8 ± 0.75 ^b^
Milk protein [%]	3.8 ± 0.40 ^a^	3.7 ± 0.47 ^b^
Milk lactose [%]	4.8 ± 0.20	4.6 ± 0.32
Somatic cell count [cells/mL]	651.4 ± 1378.0	741.9 ± 1426.6
Milk urea nitrogen [mg/dL]	18.1 ± 4.28	13.0 ± 4.70
Days in milk	297.9 ± 187.3	342.3 ± 216.2

**^#^** TMR—Total mixed ration (dry matter basis); ^a,b^ Means with different superscript letters in a row differ between breeds (*p <* 0,05).

**Table 4 animals-11-03068-t004:** Summary statistics (mean ± SD) of reproduction parameters in Holstein Friesian and Jersey cows included in this study.

Reproduction Parameters	Holstein Friesian (*n* = 2087)	Jersey (*n* = 5776)
Calving number	2.31 ± 0.75	2.2 ± 0.82
Inter calving period [days]	434.8 ± 77.7	413.3 ± 95.3
Days to conception	154.4 ± 95.9	146.9 ± 91.1
Reproduction performance [%]	65.8 ± 10.4	70.2 ± 13.7
Reproductive index	104.9 ± 2.9 ^a^	109.7 ± 1.2 ^b^

^a,b^ Means with different superscript letters in a row differ between breeds (*p <* 0.05).

**Table 5 animals-11-03068-t005:** Summary of the effects of fixed coefficients on inter calving period (ICP) in Holstein Friesian and Jersey cows.

Model Term	Coefficient	Std. Error	T	Sig.
Intercept	494.158	49.9000	9.903	0.000
Breed = Friesian	29.117	21.8868	1.330	0.183
Breed = Jersey	0			
Seasons = Autumn	−10.136	4.3784	−2.315	0.021
Seasons = Spring	−16.251	4.7984	−3.387	0.001
Seasons = Summer	−41.729	6.6025	−6.320	0.000
Seasons = Winter	0			
Parity#1	−42.236	2.8411	−14.866	0.000
Parity#2	−18.803	2.3158	−8.119	0.000
Parity#3	0			
MUN	1.032	0.2307	4.475	0.000
RDP%	−3.225	0.5393	−5.981	0.000
CP %	4.673	1.3484	3.466	0.001
ME (MJ)	−5.447	2.7924	−1.951	0.051
T_Av_	2.871	0.5502	5.218	0.000
RH_Av_	0.942	0.1242	7.583	0.000

MUN—Milk urea nitrogen; RDP%—Rumen degradable protein %; CP%—crude protein %; ME (MJ)—metabolizable energy (MJ); T_Av_—Average daily temperature; RH_Av_—Average daily relative humidity.

**Table 6 animals-11-03068-t006:** Effects of fixed coefficients on reproduction performance (RP%) of Holstein Friesian and Jersey cows.

Model Term	Coefficient	Std. Error	T	Sig.
Intercept	63.987	7.2028	8.884	0.000
Breed = Friesian	−5.034	3.5729	−1.409	0.159
Breed = Jersey	0			
Seasons = Autumn	1.039	0.6221	1.670	0.095
Seasons = Spring	1.586	0.6817	2.327	0.020
Seasons = Summer	4.907	0.9380	5.232	0.000
Seasons = Winter	0			
Parity#1	5.616	0.4037	13.914	0.000
Parity#2	2.547	0.3290	7.740	0.000
Parity#3	0			
MUN (mg/dL)	−0.093	0.0328	−2.840	0.005
RDP%	0.373	0.0766	4.864	0.000
CP%	−0.814	0.1916	−4.251	0.000
ME (MJ/kg)	0.850	0.3971	2.142	0.032
T_Av_ (°C)	−0.423	0.0782	−5.412	0.000
RH_Av_ (°C)	−0.112	0.0176	−6.364	0.000

MUN—Milk urea nitrogen; RDP%—Rumen degradable protein %; CP%—crude protein %; ME (MJ)—metabolizable energy (MJ); T_Av_—Average daily temperature; RH_Av_—Average daily relative humidity.

**Table 7 animals-11-03068-t007:** Effects of season on reproduction parameters (mean and standard error) within Holstein Friesian and Jersey cows.

Reproduction Parameters	Spring	Summer	Autumn	Winter
**Holstein Friesian**	(*n* = 2087)	(*n* = 2087)	(*n* = 2087)	(*n* = 2087)
Inter calving period (days)	425.3 (5.1) ^a^	404.0 (6.8) ^b^	430.8 (5.6) ^a^	448.7 (6.7) ^c^
Reproduction performance %	66.8 (0.79) ^a^	69.5 (0.98) ^b^	66.6 (0.86) ^a^	64.1 (0.98) ^c^
Reproduction index	105.0 (0.12)	105.0 (0.14)	104.6 (0.15)	104.9 (0.12)
**Jersey**	(*n* = 5779)	(*n* = 5779)	(*n* = 5779)	(*n* = 5779)
Inter calving period (days)	422.2 (14.8) ^a^	396.2 (15.3) ^b^	420.2 (15.0) ^a^	423.6 (15.4) ^a^
Reproduction performance %	68.4 (2.7) ^a^	71.6 (2.6) ^b^	68.9 (2.6)^b^	69.1 (2.7) ^b^
Reproduction index	109.7 (0.02)	109.7 (0.03)	109.6 (0.03)	109.7 (0.02)

^a,b,c^ Means in the same row with different superscripts differ (*p <* 0.05); SCC—Somatic cell count; MUN—Milk urea nitrogen.

## Data Availability

The data can be obtained from the National Milk Recording Scheme of the Agricultural Research Council, with permission from the South African Holstein Friesian and Jersey breed societies.

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
