# Peer review of "Effects of Milk Urea Nitrogen (MUN) and Climatological Factors on Reproduction Efficiency of Holstein Friesian and Jersey Cows in the Subtropics"

_animals, 2021, doi:10.3390/ani11113068_

Round 1

Reviewer 1 Report

This study investigated the effects of milk urea nitrogen (MUN) and climatological factors on the inter calving period (ICP), reproductive performance (RP%) and reproductive index (RI) in Holstein Friesian and Jersey cows in different seasons in the subtropics. The topic of the research is important from the point of view of agricultural economics and farming profitability.

Two breeds were studied Holstein Friesian and Jersey. The authors showed that the intercalving period (ICP) was influenced by: seasons, parity, MUN, rumen degradable (RDP) and crude proteins (CR), and average metabolisable energy (ME); the reproductive performance (RP) is influenced by climatic factors, the season, and MUN, CP and ME. There is no breed effect on ICP and RP. However, the influence of breed and parity on the reproduction index (RI) has been reported. It is worth emphasizing that all these parameters are highly influenced by nutritional factors. An analysis carried out for each breed separately showed that climatic factors had a greater effect on reproductive parameters in the dairy HF breed than in the meat Jersey breed. MUN has been shown to have a negative effect on reproductive parameters in both breeds but the Jersey breed proved to be more sensitive to MUN compared to the HF. In the Jersey breed, an effect of MUN on ICP above 12.9 mg has been reported. Also, the differences between the two breeds were noted in their sensitivity to climatic factors.

The results included in the study have great potential for use in the breeding of dairy and beef cattle in countries with subtropical conditions.

I have a few comments regarding the study.

Specifically:

  1. Please provide exact sample sizes for each breed. In the text, I found information about the total number of samples tested
  2. Subparagraph 3.3 – I suggest to change the title to emphasise that it applies to data calculated for both breeds (e.g. Reproduction parameters for both breeds)
  3. Lines 258-263 move to Discussion section
  4. I suggest checking the text in case of typos (e.g. line 352 “afdeversly”)

After minor revision I recommend the manuscript to publish.

Author Response

  1. The exact sample size for Holstein Friesland and Jersey cows in the study was included in the materials and methods sections as requested. Thank you for this suggestion as it clarifies this aspect.
  2. The title of section 3.3 was changed as recommended. Again, this recommendation is appreciated.
  3. Lines 258-263 have been moved to the discussion section as recommended. In this section an explanation was included to indicate that nutritional factors influence the reproduction parameters.
  4. In-text typo's were checked and corrected as recommended.

Reviewer 2 Report

Dear Editor

The authors should specify in which years the data were taken, and if in those years there have been extreme or extraordinary climatic situations.

It is clear that heat stress profoundly alters the metabolism of production animals. Then the question should be: is it the MUN that causes the problems, or is it the heat stress that affects the MUN and other physiological parameters not evaluated?

If we look at the title we see that it refers to Holstein Fresian and Jersey cows. How many generations of adaptation have these animals had to this type of climatic conditions?

The authors should clarify these questions.

On the other hand, the manuscript needs to perform some English revisions as follows.

Line 101: The word whith is not correct.

Line 138 and 139: It seems that the verb was does not agree with the subject. Please consider changing the verb form.

Line 244: The plural verb affect does not appear to agree with the singular subject humidity. Please consider changing the verb form for subject-verb agreement.

Line 340: The word containd is not correct.

Line 352: The word afdversely is not correct.

 Line 414: The plural verb were does not appear to agree with the singular subject performance. Please consider changing the verb form for subject-verb agreement.

 Line 444: The word temeperatures is not correct.

 Line 459: The word Acknowledegments is not correct.

Author Response

  1. The years in which the data were taken are indicated in line 92 in the materials and Methods section, and a description is provided of the climatological conditions during the period of the study.
  2. The comment about the effects of heat stress on reproduction parameters is valid and acknowledged. Not all factors that affect reproduction are included in the model and although the Mixed model analysis do provide a means to assess a number of different factors, the interactions between heat stress (climatological factors), MUN and other nutritional factors are difficult to assess, and certainly warrant further investigation. 
  3. The Holstein Friesian and Jersey cows included in the study are from diary herds that have been operating in this region for approximately 20 to 30 years,  so there has been some adaptation to the environment, although the cows are well management and fed TMR's. The inherent breed characteristics and differences are still evident from the data recorded. 
  4. The English revisions indicated were corrected, and the authors are thankful for these recommendations, namely: "with" in Line 101; verb form changed in Line 138 and 139; "contained" in Line 340; "adversely" in line 352; the plural verb was changed  in Line 414; spelling of "temperatures" corrected in Line 444; and "Acknowledgements" corrected in Line 459. The plural verb was not changed in line 244 since it refers to "temperature and humidity".
  5. Thank you for reviewing the paper in such detail and for the constructive criticism and corrections. 

Reviewer 3 Report

The evaluated paper describes the relationship between milk urea nitrogen (MUN) and some climatological factors and reproduction in two common breeds of milk cows kept in subtropics. This topic is without doubt in the scope of the journal and is current and interesting rather from practical, than scientific, point of view. The Authors attempt to make an advance to the understanding of the effect of MUN, humidity and high temperature acting together during different seasons on the reproductive efficiency. The influence of these factors on reproduction evaluated separately is relatively well known, but in contrast their interrelationships and consequences for reproductive performance have been rather weakly explored. Additionally, in the literature there is a limited information about this problem in different breeds of milk cows (Holstein Friesian, Jersey). Then, the issue studied combines already well known information with some novel aspects regarding a common seasonal effects of nutrition and climate on reproduction in two breeds of milk cows. The outcomes of this study may have several important implications for practice and may be valuable for many readers interested in improvement of fertility in milk cows. Moreover, this paper gives an advice how differently manage reproduction considering nutrition and climatological factors in Holstein Friesian and Jersey milk cows.

The general study design, number of experimental animals and methodology used are satisfactory and the entire manuscript is acceptable. The conclusions are justified by the obtained data. Because this study provides the readers with partly new data, I recommend this manuscript for publication. However, in my opinion it would be profitable for this paper to implement the following changes:

  • please, explain why do you publish currently the old data recorded in years 2006-2008?
  • the title could be better phrased, because MUN and climatological factors have not a direct effect on reproduction, but in this study design (only registration), the studied factors are rather only interrelated,
  • although, the whole paper in its current form is understandable, it is not easy to follow, due to its complexity. If the Authors could find the way how to improve this manuscript in the respect of its clarity, it would be excellent,
  • the chapters Results and Discussion are intermixed, because you discuss some problems already in Results and simultaneously, you repeat some results in Discussion. According to the rules of scientific discourse style, these chapters should be separated precisely,
  • the statement that heat stress in milk cows was noted at daily temperatures > 16°C (lines 244-245) is for me a little bit confusing, because literature data are higher – please, clear it (temperature + humidity?).
  • some fertility indices like reproductive performance (RP) and reproductive index (RI) are very sophisticated and not often to find in the literature. Please, explain why did you apply them instead of any other more useful indicators?

In conclusion, I recommend this paper for publication after the major revision.

Author Response

Thank you for the constructive criticism and valuable comments in our manuscript.

  1. The possible effects of MUN and climatological factors on tyhe reproduction of dairy cows in South Africa has not been reported before. The main reason data recorded between the years 2006 and 2008 were included is because relatively complete and accurate data were recorded for breeding values, milk production, climatological factors and complete nutritional data, which were not as well recorded in other years in the regions under investigation in this study.
  2. It is true that the effects of MUN and climatological factors on reproduction of dairy cows in this study are indirect effects. This aspect is acknowledged and an explanation pertaining to this has been included in the discussion section.
  3. Thank you for the recommendations about the clarity of the paper. We have edited some headings and rearranged certain parts to ease reading. We have also moved sections from the "Results" section to the "Discussion" section to improve the style and format of the paper.
  4. The comment about heat stress occuring > 16deg C is correct - thank you for this correction, We have checked the reference and corrected the value to 23,8deg C from the paper by Du Preez et al. Thank you!
  5. The fertility indices used include the general inter calving period, and it is true that reproduction performance and reproduction index are more complex indices. The latter two indices were included to provide a better assessment over time and parities. These parameters are often used by animal breeding and genetics experts in South Africa, who are quite involved in recommendations to the local dairy industry.
  6. Thank you for the valuable and constructive criticism and corrections, it is appreciated by the authors. 

This manuscript is a resubmission of an earlier submission. The following is a list of the peer review reports and author responses from that submission.